# Immunogenicity of an mRNA-Based COVID-19 Vaccine among Adolescents with Obesity or Liver Transplants

**DOI:** 10.3390/vaccines10111867

**Published:** 2022-11-04

**Authors:** Chomchanat Tubjaroen, Sittichoke Prachuapthunyachart, Nattakoon Potjalongsilp, Pimpayao Sodsai, Nattiya Hirankarn, Peera Jaru-Ampornpan, Voranush Chongsrisawat

**Affiliations:** 1Division of Gastroenterology and Hepatology, Department of Pediatrics, Faculty of Medicine, Chulalongkorn University and King Chulalongkorn Memorial Hospital, Bangkok 10330, Thailand; 2Center of Excellence in Immunology and Immune-Mediated Diseases, Department of Microbiology, Faculty of Medicine, Chulalongkorn University, Bangkok 10330, Thailand; 3Virology and Cell Technology Research Team, National Center for Genetic Engineering and Biotechnology (BIOTEC), Pathum Thani 12120, Thailand

**Keywords:** COVID-19, SARS-CoV-2, liver transplantation, obesity, BNT162b2 vaccine, SARS-CoV-2 variants

## Abstract

There are limited data regarding the immunogenicity of mRNA-based SARS-CoV-2 vaccine BNT162b2 among immunosuppressed or obese adolescents. We evaluated the humoral immune response in adolescents with obesity and adolescent liver transplant recipients (LTRs) after receiving two BNT162b2 doses. Sixty-eight participants (44 males; mean age 14.9 ± 1.7 years), comprising 12 LTRs, 24 obese, and 32 healthy adolescents, were enrolled. Immunogenicity was evaluated by anti-SARS-CoV-2 spike protein immunoassay and surrogate viral neutralization tests (sVNT) against the Delta and Omicron (BA.1) variants. At 27.1 ± 3.2 days after the second dose, the antibody levels were 1476.6 ± 1185.4, 2999.4 ± 1725.9, and 4960.5 ± 2644.1 IU/mL in the LTRs, obese adolescents, and controls, respectively (*p* < 0.001). Among obese individuals, liver stiffness <5.5 kPa was associated with higher antibody levels. The %inhibition of sVNT was significantly lower for the Omicron than that for the Delta variant. Injection site pain was the most common local adverse event. Nine participants (three obese and six controls) developed COVID-19 at 49 ± 11 days after the second vaccination; four were treated with favipiravir. All infections were mild, and the patients recovered without any consequences. Our study supports the need for the booster regimen in groups with an inferior immunogenic response, including LTRs and obese individuals.

## 1. Introduction

The coronavirus disease 19 (COVID-19) outbreak has become an evolving global health crisis. Since December 2019, approximately 600 million cases have been reported worldwide [1]. The BNT162b2 vaccine, containing nucleoside-modified messenger RNA (mRNA) encoding the severe acute respiratory syndrome coronavirus 2 (SARS-CoV-2) spike glycoprotein, was the first vaccine approved for use in adolescents aged 12–18 years and immunocompromised populations in May and August 2021, respectively [2,3]. Thailand’s voluntary vaccination policy for children aged 12–18 years was initiated in November 2021. Although healthy children with SARS-CoV-2 infection usually have milder symptoms than adults, an increased risk of infection susceptibility and severity has been reported among children with immune dysfunction, including those receiving immunosuppressants or living with obesity [4,5,6,7,8,9]. 

Previous studies have demonstrated that humoral immune responses to various vaccines are suppressed in solid organ transplant recipients (SOTRs) [10,11]. Various reports have revealed a diminished antibody response to the mRNA COVID-19 vaccine in adult SOTRs, particularly those of older age, those with renal impairment and other comorbidities, or those taking mycophenolate mofetil (MMF) or high-dose steroids [12,13,14,15]. A study from Israel assessed humoral antibody responses to the BNT162b2 vaccine in 80 adult liver transplant recipients (LTRs) and showed that LTRs developed lower vaccine immunogenicity than healthy controls. Antibodies were detected in only 47.5% of LTRs, in which the titers were lower than those in the healthy controls (95.4 vs. 200.5 IU/mL, *p* < 0.001) [16]. The first report on pediatric SOTRs was from the United States, and it revealed that among 19 LTRs aged 12–18 years who received the BNT162b2 vaccine, 89% developed an antibody response after two doses; none developed any serious adverse events (AEs), such as myocarditis, anaphylaxis, liver graft rejection, or death [17]. However, because of the sample size and ethical limitations, the study could not conclude the immunogenicity and safety of BNT162b2 in pediatric LTRs [17].

Childhood obesity is one of the most serious health concerns worldwide. Obesity impairs the immune system and leads to various disease, such as diabetes, cardiovascular disease, and non-alcoholic liver disease (NAFLD). Increasing data on immunogenicity to the COVID-19 vaccine in obese adults have suggested that some COVID-19 vaccines may be less effective in obese individuals [18,19,20]. In China, a multicenter study (October 2020 and February 2021) evaluated antibody responses to an alum-adjuvanted inactivated COVID-19 vaccine in 381 adults with obesity and comorbid NAFLD (males 47%, median age 39 years) and detected neutralizing antibodies (NAbs) in 95.5% of the patients [21]. However, data on obese children, especially Asian children, are limited.

Therefore, we aimed to evaluate immunogenicity among children with liver transplant (LT) and obesity following two doses of the BNT162b2 vaccine.

## 2. Materials and Methods

### 2.1. Study Design

#### 2.1.1. Study Population

This prospective study was conducted between November 2021 and May 2022. All participants were vaccinated under the voluntary vaccination policy of Thailand. The inclusion criteria for the study were patients aged 12–18 years who underwent LT at King Chulalongkorn Memorial Hospital or were diagnosed as obese according to the body mass index (BMI) using the WHO Child Growth Standards and had no known history of SARS-CoV-2 infection. Healthy controls aged 12–18 years were recruited from a high school. LTRs and obese participants were enrolled from the outpatient departments of King Chulalongkorn Memorial Hospital. All groups received the same type, dose, and interval of COVID-19 vaccination. The exclusion criteria were asymptomatic SARS-CoV-2 infection, defined as a positive result of antibody to SARS-CoV-2 and receiving live attenuated vaccines within 4 weeks of the commencement of the study. Written informed consent was obtained from all participants and their parents.

#### 2.1.2. Anthropometric Measurement and Data Collection

The serum of participants was collected approximately 4 weeks after the administration of the second vaccine. Serum samples were analyzed for SARS-CoV-2 humoral immune responses. To ensure that none of the participants had been exposed to SARS-CoV-2, we tested for the presence of antibodies to the SARS-CoV-2 nucleocapsid. Relevant data were extracted from the medical records, including demographic data, complete blood count, liver function tests, lipid profiles, immunosuppressant agents, and liver stiffness measurement (LSM) with controlled attenuation parameters (CAP) using FibroScan^®^ (Echosense, Paris, France). The examination started with the M probe on the right hepatic lobe between the 9th and 11th intercostal spaces on the midaxillary line. LSM was described in kilopascals (kPa), with the cutoff values for liver fibrosis as follows: <5.5 kPa—no fibrosis, 5.6 kPa—mild, 7.2 kPa—significant, 9.5 kPa—advanced, and 12.5 kPa—cirrhosis [22]. Liver steatosis measured by CAP was expressed in decibels/meter (dB/m), and steatosis degrees were characterized as: mild—248 dB/m, moderate—268 dB/m, and severe—280 dB/m [23].

Anthropometric parameters of all participants, including height, weight, and waist circumference (WC) were measured, and BMI (weight (kg)/height squared (m^2^)) was calculated. Participants were classified as normal weight and obese, according to the WHO Child Growth Standards. The cutoff points for WC were used to diagnose truncal obesity according to a study on children aged 6–18 years based on data from several countries in different regions [24].

### 2.2. Study Vaccine

The enrolled participants were vaccinated with two 30-µg doses of the BNT162b2 mRNA vaccine (Pfizer-BioNTech) by intramuscular injection in the deltoid muscle, 3–4 weeks apart.

#### 2.2.1. Assessment of Immunogenicity

The Elecsys^®^ anti-SARS-CoV-2 S assay (Roche Diagnostics, Rotkreuz, Switzerland), an immunoassay using the receptor-binding domain (RBD) of the S antigen, was used for total anti-SARS-CoV-2 antibody detection according to the manufacturer’s instructions. The readout was measured using a Cobas e 411 immunoassay analyzer (Roche Diagnostics). A threshold of ≥0.8 IU/mL was considered positive for anti-SARS-CoV-2. The measurement range of Elecsys is 0.4–250 IU/mL. Assessment of samples with a total antibody concentration of more than 250 IU/mL was repeated by diluting the samples with Diluent Universal (Roche).

Participants’ serum was tested for NAbs against SARS-CoV-2 variants of concern Delta and Omicron (BA.1) using an in-house surrogate viral neutralization test (sVNT) assay. Recombinant spike RBD from Delta and Omicron (BA.1) variants and recombinant human angiotensin-converting enzyme 2 (hACE2) were produced and purified from HEK293T cells. Briefly, horseradish peroxidase (HRP)-RBD was pre-incubated with test serum (at 1:10 dilution) for 1 h at 37 °C. Afterward, it was added onto an ELISA plate precoated with hACE2. The unbound HRP-RBD was washed off, and the bound RBD-ACE2 was detected by adding 3,3’,5,5’-tetramethylbenzidine (TMB) substrate and recording the absorbance value at 450 nm (OD450). Circulating NAbs against SARS-CoV-2 competitively inhibited the RBD-ACE2 interaction, resulting in lower OD450s. The percentage of inhibition (%inhibition) was calculated from the difference in OD450 between the test and control samples. The cutoff ratio for %inhibition was set at 30%.

#### 2.2.2. Safety Assessment

All participants underwent vital sign measurements before vaccination and were monitored for immediate AEs for up to 30 min after vaccination. Late AEs were recorded via phone calls on days 3, 7, and 14 after each vaccination. The participants were also encouraged to contact the study team to report any possible infection within 6 months of receiving the second dose of the vaccine. Nasopharyngeal and oropharyngeal swabs were obtained for SARS-CoV-2 reverse transcription polymerase chain reaction (RT-PCR) to confirm the diagnosis when necessary, and treatment was provided according to the standard of care.

### 2.3. Statistical Analyses

Continuous variables were reported as mean and standard deviation (SD) or as median and interquartile range (IQR). Categorical variables were described as absolute numbers, frequencies, and percentages. Accordingly, the chi-square test, Fisher’s exact test, and Mann–Whitney *U* test were employed to assess differences between categorical or continuous variables, as appropriate. Statistical significance (two-tailed) was set at *p* < 0.05. Statistical analyses were performed using IBM SPSS version 22.

## 3. Results

### 3.1. Clinical Characteristics of Participants

A total of 87 participants were vaccinated, including 17 LTRs, 30 obese adolescents, and 40 age- and sex-matched (1:1) healthy controls. Nineteen participants (18.4%) out of the total were excluded because of prior asymptomatic SARS-CoV-2 infection (*n* = 5; 5.7%), incomplete vaccination (*n* = 8; 9.2%), and refusal to participate (*n* = 6; 6.8%). The remaining 68 participants (12 LTRs, 24 obese adolescents, and 32 healthy controls) were included in the immunogenicity analysis. The mean age of enrolled subjects (44 males) was 14.9 ± 1.7 years. On 27.1 ± 3.2 days, the mean time from the second dose to immunogenicity testing, the mean antibody levels were 1476.6 ± 1185.4, 2999.4 ± 1725.9, and 4960.5 ± 2644 IU/mL in the LTRs, obese adolescents, and controls, respectively (*p* < 0.001).

Among the 12 LTRs, the mean age was 14.5 ± 1.8 years; 66.7% were males, and the mean BMI was 18.5 ± 2.8 kg/m^2^. All recipients had the first LT, and 58.3% (*n* = 7) had received LTs from a living related donor. The median (IQR) time since LT was 102 (58–160) months. The maintenance immunosuppressants included calcineurin inhibitors (50% tacrolimus, 50% cyclosporin), corticosteroids (25%), antimetabolites (33.3% MMF, 8.3% azathioprine), and sirolimus (8.3%). The median (IQR) MMF and corticosteroid doses were 13 (10–28) and 0.3 (0.2–0.5) mg/kg/day, respectively. The median (IQR) total white blood cell count and absolute neutrophil count within one month before vaccination were 5330 (3502–6636) and 2354 (1902–3225) cells/mm^3^, respectively. The clinical characteristics of the LTRs are described in Table 1.

For the obese group (*n* = 24), the mean age was 15.3 ± 1.2 years, and 25% were males. In this group, 33% (*n* = 8) were diagnosed with type 2 diabetes mellitus (T2DM), 86% (*n* = 20) presented with truncal obesity, and 67% (*n* = 16) were found to have acanthosis nigricans on physical examination. The mean BMI and WC were 36.9 ± 10.7 kg/m^2^ and 87.2 ± 15.6 cm, respectively. The mean CAP and LSM were 284.7 ± 55.7 dB/m and 6.9 ± 1.5 kPa, respectively. Most obese participants had significant fibrosis (46%); mild or no fibrosis were found in 29% and 20% of participants, respectively, and only one participant (5%) showed advanced fibrosis. Regarding hepatic steatosis (HS), 41% of obese participants had severe steatosis, whereas moderate, mild, or no steatosis were shown by 13%, 29%, and 17% of participants, respectively. There were no statistically significant differences in metabolic profile, including triglyceride, low-density lipoprotein, cholesterol, high-density lipoprotein, LSM, and CAP, between the T2DM and non-T2DM subgroups (Table 2).

For the control group (*n* = 32), the mean age was 15.4 ± 1.2 years, 56.3% were males, and the mean BMI and WC were 21.0 ± 5.2 kg/m^2^ and 63.5 ± 6.9 cm, respectively. No statistically significant differences were observed in age, sex, height, or time from the second vaccination to immunogenicity testing between the groups.

### 3.2. Postvaccination Immunogenicity

The mean time from the second dose to immunogenicity testing was 27.1 ± 3.2 days (26.2 ± 1.4, 28.7 ± 2.2, and 27.5 ± 1.8 days for the LTRs, obese adolescents, and controls, respectively; *p* = 0.76). The antibody levels were 1476.6 ± 1185.4, 2999.4 ± 1725.9, and 4960.5 ± 2644.1 IU/mL, and %inhibition of sVNT to Delta variant were 91.2 ± 13.4, 96.6 ± 8.6, and 98.5 ± 1.6% in the LTRs, obese adolescents, and controls, respectively (Figure 1). The antibody level did not differ between LTRs taking MMF and other immunosuppressants (843.7 [440.5, 2769.8] vs. 1357.5 [493.7, 2300.7] IU/mL, *p* = 0.83). In the obese adolescent group, the severity of steatosis, comorbid T2DM, truncal obesity, and acanthosis nigricans did not affect antibody levels. Liver stiffness < 5.5 kPa was significantly associated with higher antibody levels (Table 3). The comparison of sVNT between the Delta and Omicron (BA.1) variants is presented in Figure 2.

### 3.3. Safety and Adverse Events

The incidences of AEs are shown in Figure 3. No severe AEs were observed within 30 min of each vaccination. However, within 72 h of the first dose, 59 (86.7%) participants reported a total of 97 AEs, and injection site pain was the most common (96%) AE in these participants. Other AEs included fever and chills (27%), headache (12%), myalgia (12%), fatigue (8%), and diarrhea (6%). For the second dose, 60 (89%) participants reported a total of 107 AEs, including injection site pain (90%), myalgia (21%), headache (18%), fever and chills (17%), diarrhea (6%), and fatigue (3%). On day seven after each vaccination, none of the participants reported AEs. In the LTR group, no acute cellular rejection occurred, and no significant alteration in aspartate aminotransferase (30 (23–61.5) vs. 29.5 (24.5–57) IU/mL, *p* = 0.75) and alanine aminotransferase (35 (21–112.5) vs. 28.5 (20–84.5) IU/mL, *p* = 0.53) levels was observed. All AEs were mild and resolved within 48 h. No statistically significant correlation was observed between AEs and the level of immunogenic response to the vaccine.

### 3.4. Efficacy of the Vaccine

Of the sixty-eight fully vaccinated participants, three obese adolescents and six controls developed SARS-CoV-2 infection. All patients were mildly symptomatic and were diagnosed using nasopharyngeal RT-PCR. The mean time from the date of the second vaccination was 49 ± 11 days. Two patients with COVID-19 (22.2%) were hospitalized and treated with favipiravir (*n* = 1) and *Andrographis paniculata* (Burm.f.) Nees (*n* = 1). The rest (*n* = 7; 77.8%) were managed as outpatients with favipiravir (*n* = 3), *A. paniculata* (*n* = 2), or *Boesenbergia rotunda* (L.) Mansf. (*n* = 1) treatments. The mean duration of the disease was 3.8 ± 1.1 days. All the patients recovered without disease progression.

## 4. Discussion

### 4.1. LTRs Show Low Antibody Response to the BNT162b2 Vaccine

The current study revealed that the immunogenicity of the BNT162b2 mRNA vaccine (Pfizer-BioNTech) in Thai adolescent LTRs is lower than that in healthy adolescents. Overall, 91% of LTRs showed sVNT ≥ 80% inhibition against Delta strains up to a month after vaccination. In contrast, approximately 60% of the immunocompromised Thai adolescents, including 19 hematopoietic stem cell transplantation patients (HSCT), 23 kidney transplant recipients, 15 cancer patients undergoing chemotherapy, and patients taking immunosuppressive drugs, particularly systemic lupus erythematous patients (*n* = 43), achieved sVNT ≥80% inhibition after 2 doses of BNT162b2 [25]. However, the study of Chantasrisawad et al. [25] on the Thai adolescent population did not include LTRs. More intense and higher doses of immunosuppressive agents, including methotrexate (28%), MMF (45%), and prednisolone (73%), and a shorter time between transplantation and vaccination (median 37 months after HSCT and 35 months after kidney transplant) might be correlated with a lower antibody response than that observed in our study on LTRs (median 102 months after LT) [25]. Among adult LTRs, studies have revealed that the factors associated with negative antibody responses are age, renal function, and type of immunosuppression [16,26]. Rabinowich et al. [16] reported significantly low serological response to BNT162b2 in adult LTRs (age 60.1 ± 12.8 years), with 47.5% achieving a positive NAb titer. MMF or high-dose corticosteroid treatment within one year before vaccination was correlated with a lower immunogenicity [15]. MMF has been shown to be associated with a low immune response because it interferes with T and B cell proliferation, leading to impaired antibody generation [27,28]. However, to date, LTRs show better results than SOTRs [15]. Although a low antibody response was observed in the LTR group, there was no breakthrough infection for 6 months after vaccination. This could be because of strict preventive behaviors, including social distancing, wearing masks, and personal hygiene.

### 4.2. Obese Adolescents Show Low Antibody Response to the BNT162b2 Vaccine

There are increasing data on the impact of obesity on responses to vaccines, especially influenza, hepatitis B, and COVID-19 vaccines in adults, with conflicting results [29,30,31,32]. Moreover, only a few studies have reported the vaccine response in obese children and adolescents [33,34,35]. A cohort study [33] enrolled 34 obese children to evaluate monovalent influenza vaccine antibody responses and revealed no differences in seroconversion by BMI classification. Similar findings have been observed in another study [34] on a trivalent influenza vaccine that included 23 children of normal weight and 28 obese children aged 3–14 years. A recent study [35] on antibody response to live attenuated hepatitis A vaccine in 95 obese children compared with 117 normal-weight children revealed a 100% seroprotection rate with no difference in postvaccination geometric mean titers between each group. However, only limited data on COVID-19 vaccine responses in obese children are available. In a phase III trial of the BNT162b2 vaccine involving 6556 obese participants, the vaccine efficacy one week after the second dose of the vaccine did not differ between the obese (95.4%) and normal-weight (94.8%) groups [36]. When stratified by age, there were no significant differences in vaccine efficacy, which was 95% in the young obese adults aged 16–24 years [36]. In this study, SARS-CoV-2 antibody levels at approximately 4 weeks following the second dose of the BNT162b2 vaccine in obese adolescents were lower than those in the control participants with normal weight. However, no difference in the seroprotective rate was observed.

Obese people are hypothesized to be predisposed to poor immunological responses to various vaccinations due to their chronic inflammatory state and immune dysregulation as well as other related comorbidities, including NAFLD and T2DM. Individuals with truncal obesity may be predisposed to an increased inflammatory state compared to those without truncal obesity. This is because the abnormal accumulation of adipose tissue within the abdomen can lead to the production of adipokines and proinflammatory cytokines, which influences T and B cell immune responses. Adipose tissue not only acts as a lipid storage and source of energy, but is also an endocrine organ secreting fatty acids, metabolites, and adipokines [37]. These adipokines play an important role in inflammation, immune response, and regulation of glucose metabolism. Leptin and adiponectin are crucial adipokines that are in a state of imbalance in obesity [37,38,39]. Leptin’s proinflammatory role influences the adaptive arm of T and B cells immune responses [40]. In contrast, adiponectin plays an anti-inflammatory role, suppressing T cell-mediated and innate immune responses. An imbalance of adiponectin and leptin in obesity leads to a low-grade inflammatory status [41]. Furthermore, the role of truncal obesity in immune responses to vaccination is inconclusive. Watanabe et al. revealed that adults with truncal obesity and a higher WC had significantly lower SARS-CoV-2 antibody titers [18]. However, this study found no significant difference in antibody levels between obese adolescents with and without truncal obesity. In contrast, the previous study [35] on hepatitis A revealed that subjects with truncal obesity mounted a significantly higher antibody response to hepatitis A virus than the rest of the study population.

The incidence of NAFLD in obese populations is rising, and the data on the immunological responses to COVID-19 vaccines in adult patients with NAFLD are increasing. However, limited data are available on the impact of NAFLD on COVID-19 vaccine responses in pediatric populations. A multicenter study conducted in China on COVID-19 vaccine immunogenicity and safety in patients with NAFLD (*n* = 381; median age, 39 years; BMI, 26.1 kg/m^2^) demonstrated the effectiveness and safety profile of the vaccine [21]. Furthermore, Cheung et al. [42] studied the effect of moderate to severe HS on the BNT162b2 vaccine immunogenicity in 68 patients and 168 controls with a median age of 51 years. They demonstrated no differences in the seroconversion rate between moderate/severe HS and the control group, but a lower proportion of patients with NAFLD achieved the highest titer of humoral responses after two doses of vaccines compared to the healthy controls. The present study included subgroup analysis on HS severity; it revealed no difference in antibody levels after BNT162b2 administration between the “none–mild HS” and “moderate–severe HS” in patients with NAFLD. Adaptive immune dysfunction in cirrhotic patients leads to hypo-responsiveness to vaccines [43]. The current study supports that statement and shows that the patients with LSM ≥ 5.5 kPa were associated with poor antibody development. We suspected that obese patients with LSM ≥ 5.5 kPa may have a more advanced liver disease than those with LSM < 5.5 kPa. Taken together, we inferred that the patients with NAFLD with higher LSM may be in the progression phase of chronic liver disease, which greatly alters immune responses.

Furthermore, obese patients also present with hyperinsulinemia, which can aggravate impaired immune function. Insulin has an important role in T cell function, promoting anti-inflammatory T helper 2 cell response [44], whereas insulin resistance is associated with proinflammatory T helper 1 cell response [45]. However, it is uncertain whether comorbid T2DM alters vaccine immunogenicity. Moreover, in the current study, no significant difference was observed in vaccine immunogenicity between the T2DM and non-T2DM groups. Consistent with the previous reports [46,47] that examined the humoral immune response in adults with T2DM, it was found that antibody levels after the second dose of mRNA COVID-19 vaccine (BNT162b2 and mRNA-1273) were similar between participants with T2DM and healthy controls. Correspondingly, an Italian study involving 150 participants revealed that comorbid T2DM and hyperglycemia did not affect the production of NAbs [47]. In contrast, the CAVEAT study found that NAbs and CD4 cytokine responses were lower in patients with T2DM with HbA1c levels > 7% than those with HbA1c levels ≤ 7%, implying that hyperglycemia at the time of vaccination might worsen the immunological response [48]. A recent systematic review in adults emphasized that the seroconversion rates after SARS-CoV-2 vaccination in T2DM patients were lower than those in the healthy controls [49]. The inconclusive results may be due to the differences in age, renal function, glycemic control, and other comorbidities among the studies.

### 4.3. Safety

In our study, we observed that injection site pain was the most common local AE. It occurred in 90–96% of participants after each dose, which was higher than the previous reports on adolescents at 73–86% [25,50]. The variations in reaction rate could be because of differences in participant demographics, including age, weight, sex, and comorbidities. The common systemic reactions were fever, myalgia, and headache, occurring in one-third of participants in the pivotal trial of BNT162b2 among the adolescent group [50]. Similar to other studies [51,52], we could not find an association between the humoral response levels and local or systemic reactions in our participants. Furthermore, there are several concerns about SARS-CoV-2 vaccines in LTRs, such as inducing graft rejection or a viral-specific immune response. Our study showed no serious AE in LTRs receiving either dose of BNT162b2.

Our study has some limitations. First, we included a small number of LTRs, which is statistically inadequate to identify the significant impact factor of the lower humoral response. Second, we collected the data during the Delta strain wave, and 2 months later, the Omicron strain became predominant. Consequently, breakthrough infections occurred, and the two doses of BNT162b2 could not protect against infection. Third, we did not measure T cell response after vaccination, which could determine if participants with a low humoral immune response could develop a sufficient T cell response to prevent COVID-19. Detailed immunological studies, including characterization of T cell and memory B cell responses, may be essential in determining further vaccination regimens. Fourth, in the obese group, we did not perform a liver biopsy to determine the degree of liver fibrosis and correlate with LSM. Though LSM is a surrogate biomarker of liver fibrosis, hepatic steatosis is one of the confounding factors affecting the LSM value. A previous report revealed that healthy children with hepatic steatosis, as shown by ultrasound, had greater LSM than those without steatosis [53].

## 5. Conclusions

This study shows that adolescents with LT and obesity developed lower immunogenicity with two doses of the BNT162b2 vaccine. They had a high %inhibition of sVNT to Delta variant up to 1 month after vaccination. However, %inhibition of sVNT to Omicron (BA.1) is significantly lower than to the Delta variant, supporting the administration of a booster dose in these groups.

## Figures and Tables

**Figure 1 vaccines-10-01867-f001:**
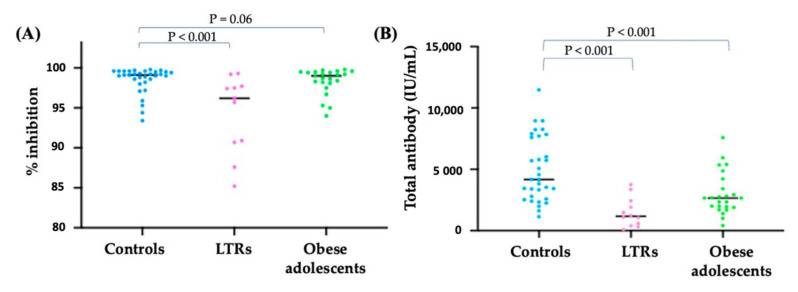
(**A**) The sVNT (%inhibition) to Delta variant (**B**) The SARS-CoV-2 total antibody after second dose of vaccination in healthy controls, LTRs, and obese adolescents.

**Figure 2 vaccines-10-01867-f002:**
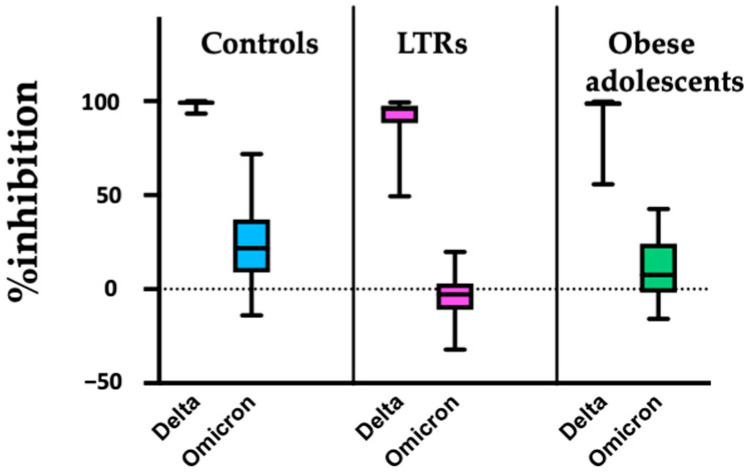
Comparison of sVNT (%inhibition) between Delta and Omicron (BA.1) variants after administering the second dose of BNT162b2 in healthy controls, LTRs, and obese adolescents.

**Figure 3 vaccines-10-01867-f003:**
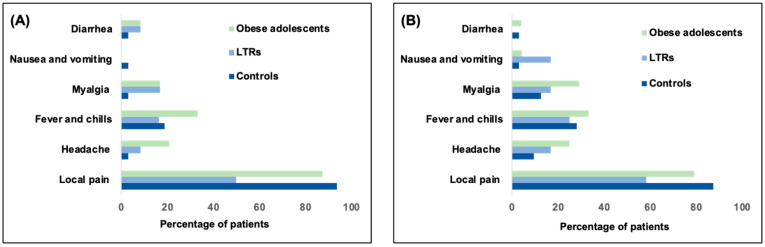
Development of adverse events after the first (**A**) and second (**B**) dose of vaccination within 7 days.

**Table 1 vaccines-10-01867-t001:** Characteristics of liver transplant recipients.

Characteristics	N = 12
Age, year (mean ± SD)	14.5 ± 1.8
Males, *n* (percentage)	8 (66)
BMI, kg/m^2^ (mean ± SD)	18.5 ± 2.8
Primary disease, *n* (percentage)	
Biliary atresia	7 (58.4)
Autoimmune liver disease	1 (8.3)
Alagille syndrome	1 (8.3)
Others	3 (25)
Time from transplantation to vaccination, months (median (IQR))	102 (58–160)
Immunosuppressants, *n* (percentage)	
Corticosteroid	3 (25)
Calcineurin inhibitor	
• Cyclosporin	6 (50)
• Tacrolimus	6 (50)
Antimetabolites	
• MMF	4 (33.3)
• Azathioprine	1 (8.3)
mTOR inhibitor (sirolimus)	1 (8.3)
Number of immunosuppressants, *n* (percentage)	
1	6 (50)
2–3	6 (50)

BMI, body mass index.

**Table 2 vaccines-10-01867-t002:** Baseline characteristics of the obese group with or without T2DM comorbidity.

Parameters	T2DM (*n* = 8)	Non-T2DM (*n* = 16)	*p*-Value
Triglyceride (mg/dL)	157.88 ± 47.83	143.19 ± 65.42	0.58
Low-density lipoprotein (mg/dL)	136.5 ± 32.94	127.81 ± 35.16	0.56
Cholesterol (mg/dL)	165.63 ± 22.6	176.31 ± 31.58	0.40
High-density lipoprotein (mg/dL)	48.75 ± 9.54	44.56 ± 10.33	0.34
LSM (kPa)	7.25 ± 1.55	6.81 ± 1.55	0.52
Liver stiffness stage (*n*, %)			
None–mild	3 (37.5%)	10 (62.5%)	0.44
Significant–advanced	5 (62.5%)	6 (37.5%)	0.23
CAP (dB/m)	289.25 ± 55.42	282.44 ± 57.55	0.78
Steatosis stage (*n*, %)			
None–mild	3 (37.5%)	8 (50%)	0.94
Moderate–severe	5 (62.5%)	8 (50%)	0.87

Data expressed as mean ± SD. LSM, liver stiffness measurement; CAP, controlled attenuation parameter.

**Table 3 vaccines-10-01867-t003:** Association between baseline characteristics and postvaccination antibody values in the obese adolescent group.

Factors (*n*)	Antibody Levels (IU/mL)	*p*-Value
Steatosis		
-None–mild (11)	2342 (1759–3834.7)	0.79
-Moderate–severe (13)	2772.5 (1914.3–4513.5)	
Liver stiffness measurement		
-LSM < 5.5 kPa (5)	4228 (2165–6762.5)	0.03
-LSM ≥ 5.5 kPa (19)	2344 (1092–2924)	
Comorbid T2DM		
-yes (8)	2801.5 (2001.2–4870.5)	0.59
-no (16)	2497.5 (1684.7–3871.3)	
Truncal obesity		
-yes (20)	2667 (1914.5–4027.5)	0.09
-no (4)	2496 (1336–6350.75)	
Acanthosis nigricans		
-yes (16)	2699 (1914.5–4027.5)	0.69
-no (8)	2496 (1759.5–5125.5)	

Data expressed as median (IQR).

## Data Availability

Not applicable.

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
