# Peer review of "Immunogenicity of an mRNA-Based COVID-19 Vaccine among Adolescents with Obesity or Liver Transplants"

_vaccines, 2022, doi:10.3390/vaccines10111867_

Round 1

Reviewer 1 Report

In this study, the authors evaluated the humoral immune response and presented the adverse event in adolescents with obesity or liver transplant recipients (LTRs) after receiving two BNT162b2 doses. The findings have some scientific contributions. However, a number of issues should be addressed before acceptation for publication.

Specific comments:

1.     The authors only tested the humoral immune responses at 27.1 ± 3.2 days after the second dose. How about 1 to 3 weeks after the first and second doses?

2.     A temperature curve for vaccinated participants after vaccination could improve the manuscript.

3.     A neutralizing antibody titer curve after vaccination could improve the manuscript.

4.     Divide the discussion part with separate topics and titles will make it more understandable for the audience.

Author Response

Dear Reviewers,

We appreciate the time and effort you have dedicated to providing insightful feedback on ways to strengthen our manuscript. Below we have provided point-by-point responses to all your raised concerns.

  1. We have revised the introduction section to provide sufficient background for the study. In addition, we have also made pertinent changes in other sections of the manuscript to ensure better understandability.
  2. The authors only tested the humoral immune responses at 27.1 ± 3.2 days after the second dose. How about 1 to 3 weeks after the first and second doses?

    Response: Thank you for raising this question. We agree that these data could have further strengthened our results. However, given the constraints of traveling and cost, we could not perform the blood test at 1st and 3rd week of each dose in normal controls [they lived in urban areas of Thailand, and it takes around 5–8 h from the hospital to reach there]. In LTRs, the blood tests were performed coinciding with their routine follow-up visits at which the patient felt comfortable and had good compliance. In the case of obese participants, some required 2–3 attempts to draw the blood sample because of the arm’s fat.

    Given these limitations, we decided not to perform repeated blood tests, as patient care and safety were also important.

    3.  A temperature curve for vaccinated participants after vaccination could improve the manuscript.

    Response: Thank you for this comment. We agree that providing information on the temperature curve of the participants post-vaccination could have provided additional data to strengthen our study. Unfortunately, we could not develop a temperature curve, as the data were collected by phone and we were not sure of the accuracy of individual temperature measurements.

    4.     A neutralizing antibody titer curve after vaccination could improve the manuscript.

    Response: Thank you for this comment. We could not develop a titer curve, as data of NAbs were performed as %inhibition. 

         5. Divide the discussion part into separate topics and titles will make it more understandable for the audience.

         Response: Thank you for this suggestion. Accordingly, we have divided the discussion section into different headings to highlight the immunogenicity of the vaccine in each group.

Reviewer 2 Report

The authors aimed to evaluate immunogenicity among children with liver transplant 63 (LT) and obesity following two doses of the BNT162b2 vaccine.

The study covers some issues that have been overlooked in other similar topics. The structure of the manuscript appears adequate and well divided in the sections. Moreover, the study is easy to follow, but some issues should be improved. Some of the comments that would improve the overall quality of the study are:

a. Authors must pay attention to the technical terms acronyms they used in the text.

b. Conclusion Section: This paragraph required a general revision to eliminate redundant sentences and to add some "take-home message".

Author Response

Dear Reviewers,

We appreciate the time and effort you have dedicated to providing insightful feedback on ways to strengthen our manuscript. Below we have provided point-by-point responses to all your raised concerns and made pertinent changes in the manuscript as suggested. We hope the responses and the changes are satisfactory and look forward to your positive feedback for the publication of our manuscript. We are open to further discussion that you think can improve our manuscript.

  1. Dear Reviewer, thank you for highlighting this. We have carefully checked all technical terms and ensured that all acronyms are defined appropriately.

  2. Thank you for your in-depth review. We apologize for not being clear. We have revised the conclusion section to avoid redundancy and incorporated a take-home message.
